# Cooperative supramolecular polymerization of styrylpyrenes for color-dependent circularly polarized luminescence and photocycloaddition

Wei Yuan[1,5], Letian Chen[2,5], Chuting Yuan[3,5], Zidan Zhang [4], Xiaokai Chen[1], Xiaodong Zhang [1], Jingjing Guo[1], Cheng Qian [1], Zujin Zhao [2] & Yanli Zhao [1]

Developing facile and efficient methods to obtain circularly polarized luminescence (CPL) materials with a large luminescence dissymmetry factor ($g_{lum}$) and fluorescence quantum yield ($\Phi_Y$) is attractive but still challenging. Herein, supramolecular polymerization of styrylpyrenes ($R$/$S$-PEB) is utilized to attain this aim, which can self-assemble into helical nanoribbons. Benefiting from the dominant CH·π interactions between the chromophores, the supramolecular solution of $S$-PEB shows remarkable blue-color CPL property ($g_{lum}$: 0.011, $\Phi_Y$: 69%). From supramolecular solution to gel, the emission color (blue to yellow-green) and handedness of CPL ($g_{lum}$: −0.011 to +0.005) are concurrently manipulated, while the corresponding supramolecular chirality maintains unchanged, representing the rare example of color-dependent CPL materials. Thanks to the supramolecular confine effect, the [2 + 2] cycloaddition reaction rate of the supramolecular solution is 10.5 times higher than that of the monomeric solution. In contrast, no cycloaddition reaction occurs for the gel and assembled solid samples. Our findings provide a vision for fabricating multi-modal and high-performance CPL-active materials, paving the way for the development of advanced photo-responsive chiral systems.

The construction of circularly polarized luminescent (CPL) materials has gained enormous attention owing to their promising applications in 3D display[1–3], optical sensors[4–6], and photoelectric devices[7–9]. An ideal CPL material requires multi-color emission, high photoluminescence quantum yield ($\Phi_Y$), and large luminescence dissymmetry factor ($g_{lum}$)[10]. However, enlargement of $g_{lum}$ and $\Phi_Y$ often requires at the expense of each other. Recently, several co-assembly methods accompanied by energy transfer have been utilized to construct multi-color CPL materials with high $g_{lum}$ and $\Phi_Y$[11–15]. However, the co-assembly strategy involves a tedious operation process. In addition, the co-assembly process is often accompanied by self-aggregation and self-sorting[16], which brings batch-to-batch discrepancies in CPL properties. Therefore, seeking an easy strategy to realize multi-color and high-performance CPL properties is desired.

With these issues in mind, our motivation was to construct CPL-active materials through supramolecular polymerization of one-

[1]School of Chemistry, Chemical Engineering and Biotechnology, Nanyang Technological University, 21 Nanyang Link, Singapore 637371, Singapore. [2]State Key Laboratory of Luminescent Materials and Devices, Guangdong Provincial Key Laboratory of Luminescence from Molecular Aggregates, South China University of Technology, Guangzhou 510640, China. [3]CAS Key Laboratory of Soft Matter Chemistry, Department of Polymer Science and Engineering, University of Science and Technology of China, Hefei, Anhui 230026, China. [4]McKetta Department of Chemical Engineering, University of Texas at Austin, Austin, TX 78712, USA. [5]These authors contributed equally: Wei Yuan, Letian Chen, Chuting Yuan. ✉e-mail: mszjzhao@scut.edu.cn; zhaoyanli@ntu.edu.sg

component chiral chromophore. To date, one-component CPL-active materials with multi-color property have been rarely reported[17,18], due to the difficulty of combining chirality and strong luminescence. Typically, strong intermolecular interactions, such as π-π interactions, hydrogen bonding, halogen bonding etc[19–22], are required to obtain helical architectures such as helix nanobelts and nanotubes in the assembly process. However, strong π-π interactions between chromophores can seriously reduce luminescence property[23]. In order to obtain one-component CPL material with high $\Phi_Y$, the π-π interactions between chromophore moieties must be suppressed on the premise of maintaining supramolecular chirality. CH-π interaction[24], unconventional hydrogen bonding, is a kind of weak intermolecular or intramolecular force between the H atom and the π-conjugated plane[25], which plays important roles in controlling crystal packing and molecular recognition processes[26,27]. In general, the strength of CH-π interaction between aromatic compounds is stronger than that of the aliphatic ones. These aromatic CHs favor the edge-to-face or T-shape π/π stacking, which helps to inhibit the quenching of fluorescence[28]. Another distinguishing feature of CH-π interaction is their cooperative work manner, which is extensively demonstrated in crystallography. Therefore, we employ CH-π interactions as the dominant force between chromophores to fabricate chiral supramolecular polymers with boosting CPL properties. For the one-component system, multi-color and multi-modal CPL properties are important for practical applications. Multi-color emission can be realized by varying stacking states[29,30]. The chirality of supramolecular materials can be controlled by assembly methods[31,32], thereby manipulating the handedness of CPL[33]. Therefore, achieving multi-color and multi-modal CPL properties with high performance can be expected via the supramolecular polymerization of a one-component system.

Pyrene has attracted widespread attention due to outstanding photostability, high luminescence quantum yield, and modifiability[34]. Styrylpyrene is one of the pyrene derivatives with C = C as a linker, which has been utilized in fluorescence probes and biological material[35]. Although the fluorescence property of styrylpyrene has been widely investigated, the corresponding helical nanostructures with efficient CPL property have been limited to date. It is found that styrylpyrenes are prone to stack in the head-to-tail "antiparallel" mode driven by multiple CH-π interactions in crystalline state[36]. Such arrangement behavior exhibits the absence of π-π interactions between molecular planes, which ensures strong luminescent properties in aggregated states. When chiral units are attached to styrylpyrene core, helical supramolecular polymers with strong emission are obtained through supramolecular polymerization[37], which allows for CPL property. Since the C = C bond is a classical photochromic group, and photoreaction of styrylpyrene has been studied in dilute solution and crystalline state[38,39]. Compared with dilute solution and crystal, the supramolecular solution combines mobility and aggregation owing to dynamic non-covalent interactions, which may possess unique photoresponsive behavior[40–43]. However, the photo-responsive behavior of helical supramolecular polymers of styrylpyrenes has not been explored.

In this regard, we synthesize two chiral styrylpyrene derivatives (S-PEB and R-PEB; Supplementary Fig. 1) and study their supramolecular polymerization, CPL behavior of various aggregates, and photo-responsive properties (Fig. 1a). The optically active units S or R-(N-(1-phenylethyl) group) are close to the styrylpyrene core, which facilitate the chirality transfer at the supramolecular level via supramolecular polymerization. It is found that both S-PEB and R-PEB can form long-range ordered spiral nanostructures driven by CH-π interactions. Four self-assembled materials (supramolecular solution, gel, drop-casting film, and co-assembled PMMA film) all show high-performance CPL properties ($g_{lum} = 1 \times 10^{-3} - 1.1 \times 10^{-2}$ and $\Phi_Y = 28\% - 69\%$) (Fig. 1b). Intriguingly, not only the emission color but the handedness of CPL is manipulated for various assembled states, presenting a unique example of realizing color-dependent CPL properties in a single-component system. Benefiting from supramolecular confinement effect, we observe that the photo-triggered [2 + 2] cycloaddition reaction rate of the S-PEB supramolecular solution is 10.5 times higher than its monomeric solution (Fig. 1c). Interestingly, there is no occurrence of cycloaddition reaction in supramolecular gel or self-assembled solid, which is possibly due to the restriction of intermolecular mobility. The present work thus provides a simple strategy for developing multi-modal and high-performance CPL materials and chiral photo-responsive smart devices.

## Results

The supramolecular self-assembly of S-PEB was firstly investigated through UV-vis and circular dichroism (CD) spectroscopy. In dilute 1, 1, 2, 2-tetrachloroethane (TCE, $C = 4 \times 10^{-4}$ M) solution, the maximum absorption bands of S-PEB were located at 314 nm and 389 nm, respectively (Fig. 2a). By contrast, switching the solvent from pure TCE to methylcyclohexane/1,1,2,2-tetrachloroethane solvent (MCH/TCE = 24:1, v/v), the maximum absorption bands were shifted to 304 nm, and 370 nm with two new shoulder peaks emerging at 328 nm and 432 nm, respectively, which indicated the occurrence of self-assembly behavior. In addition, the concentration-dependent UV-vis spectra and dynamic light scattering (DLS) measurements further confirmed the presence of aggregates in MCH/TCE solution (Supplementary Figs. 2, 3). Upon increasing the temperature of the mixed solution of S-PEB (Supplementary Fig. 4a), two isosbestic points ($\lambda = 335$ and 420 nm) were observed, validating the reversible transition between the monomeric and aggregated states[44]. In addition, CD spectroscopy was used to investigate the supramolecular chirality of S-PEB aggregates. As shown in Fig. 2b, bisignate CD signals were shown in the mixed solvent of MCH/TCE, suggesting the formation of long-range ordered helical aggregates. There are three negative CD signals at 356 nm ($g_{abs} = -0.0038$), 333 nm ($g_{abs} = -0.0032$), 290 nm ($g_{abs} = -0.0016$), and two positive CD signals at 392 nm ($g_{abs} = 0.0071$), 310 nm ($g_{abs} = 0.0030$). By contrast, no Cotton effect was found in the TCE solution.

Hence, the chirality transfer was successfully achieved from alkyl carbon to the pyrene core via supramolecular polymerization[45]. Upon increasing the temperature, the CD signals decreased continuously and became silent over 333 K, manifesting the dissociation of helical supramolecular polymers (Supplementary Fig. 4b). R-PEB exhibited the mirror image CD signals and a similar trend in varied-temperature CD spectra as S-PEB (Supplementary Fig. 5). To further understand the supramolecular self-assembled behaviors of R-PEB and S-PEB, transmission electron microscopy (TEM) was then performed to investigate the morphology of aggregates obtained from their MCH/TCE solution. As depicted in Fig. 2c and Supplementary Fig. 6a, long-range ordered M-type helical nanobelts with nanometer size were visualized for the S-PEB supramolecular polymers. The R-PEB enables to self-assemble into P-type helical nanoribbons (Fig. 2d and Supplementary Fig. 6b). Hence, supramolecular polymers of R-PEB and S-PEB tend to bundle with each other, thereby forming an ordered helix fiber network after aging for several hours.

To study insights into supramolecular polymerization, the varied-temperature CD spectroscopy of S-PEB ($2 \times 10^{-4}$ M) was performed. By monitoring the CD signals at 387 nm with temperature-dependent spectral measurements, a non-sigmoidal heating curve with a critical temperature point ($Te$: 314 K) was presented, suggesting the nucleation-elongation cooperative mechanism (Fig. 3a)[19,44]. The thermodynamic parameters for cooperative supramolecular polymerization were obtained via the van't Hoff plot (Supplementary Fig. 7). The enthalpy (ΔH) and entropy (ΔS) for the whole self-assembled process were determined to be −88.8 kJ mol$^{-1}$ and −211.9 J mol$^{-1}$ K$^{-1}$, respectively, suggesting the enthalpy-driven process for supramolecular polymerization of S-PEB.

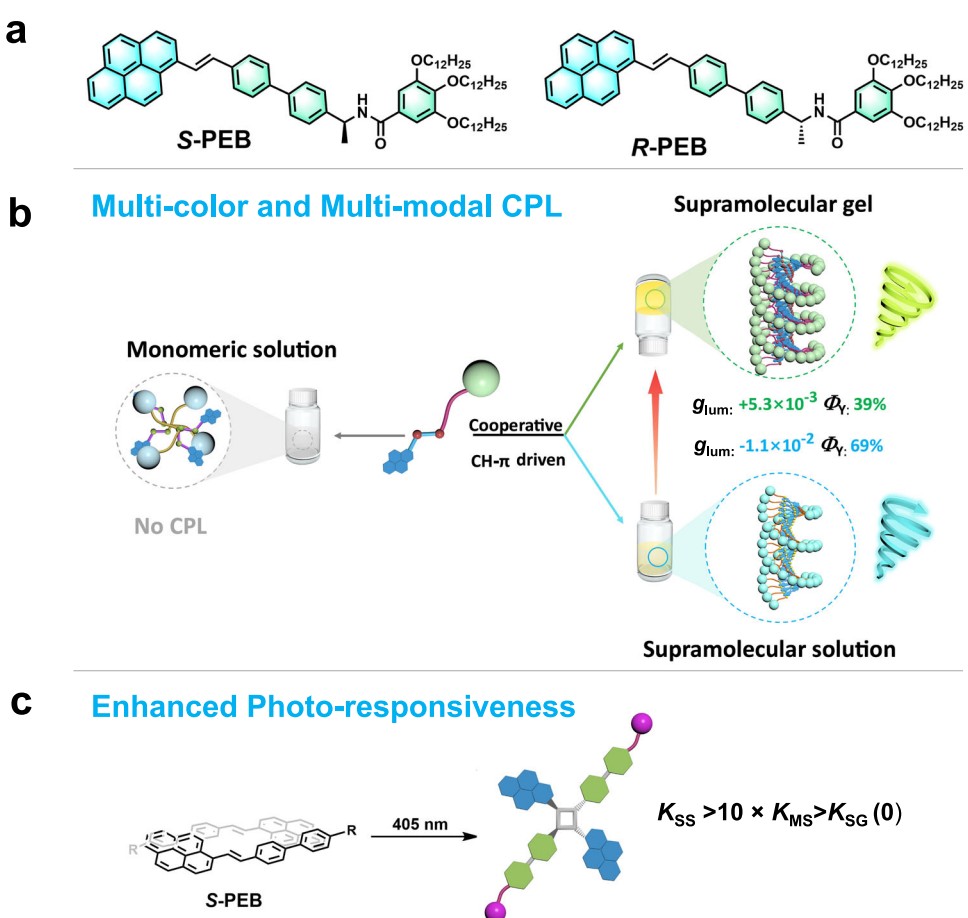

**Fig. 1 | Molecular structures, CPL properties and [2 + 2] cycloaddition.**
**a** Molecular structures of *S*-PEB and *R*-PEB. **b** Multi-color and multi-modal CPL properties of the *S*-PEB in various assembled states. **c** Different [2 + 2] cycloaddition reaction rates *K*. $K_{SS}$ for the supramolecular solution, $K_{MS}$ for the monomeric solution, and $K_{SG}$ for the supramolecular gel.

To clarify the dominant interactions between the styrylpyrenes, the concentration-dependent ${}^1H$ NMR experiment was performed. NMR signals of the NH proton (6.2 ppm) and aromatic core (8.4–7.4 ppm) both remained unchanged upon increasing the concentration (Supplementary Fig. 8). Therefore, it implies that neither hydrogen bonding nor π-π stacking is the main interactions for the self-assembly of *S*-PEB[34]. In addition, since fluorescence (FL) spectra of the *S*-PEB in the aggregated (MCH/TCE solution) and monomeric states (TCE solution) are essentially identical (Fig. 3b), it can be presumed that the intermolecular π-π distance of pyrene units is large, which leading to monomeric emission of pyrene in the supramolecular solution of *S*-PEB. The ${}^1H$-${}^1H$ COSY and NOESY NMR experiments were employed to reveal the interactions among the protons. From the 2D COSY and NOESY spectra of S-PEB with high concentration (Supplementary Figs. 9, 10), it was found that the cross peak of protons 11' and 2' existed, indicating the twist-antiparallel arrangement. DFT calculations based on the reduced density gradient (RDG) method[46] and the independent gradient model based on Hirshfeld partition (IGMH) method[47] were performed to explore the main driving force for constructing helical supramolecular polymers of *S*-PEB through Multiwfn package[48] (Fig. 3c). The optimized geometry of the dimeric stacks for *S*-PEB showed a twist-antiparallel arrangement (Fig. 3d and Supplementary Fig. 11), and neither π-π stacking nor hydrogen bond was interactionaexisted in the dimeric stacks, whereas CH-π interactions were the dominant interaction between the *S*-PEB units in the helical supramolecular polymers. CH (phenyl group) - π (center of pyrene) contacts are referred to as the edge-to-face stacking mode, which effectively inhibits the quenching effect of π-electrons. Therefore,

these DFT results were in good agreement with the experimental data (NMR and FL spectra). Previous supramolecular assemblies formed in a cooperative manner primarily rely on hydrogen bonding interactions or π-π interactions[20,49]. Although the CH-π interaction is common in crystal stacking, the supramolecular helical nanostructures driven by CH-π contact have rarely been studied. Notably, our current system is a rare example that the weaker CH-π interaction is considered as the main interaction to construct chiral supramolecular structures.

Benefiting from the absence of strong π-π interactions in the system, thus *S*-PEB and *R*-PEB possess outstanding fluorescence properties in their aggregated states. In the molecule-dissolved state (TCE solution), the maximum fluorescence peak of *S*-PEB is located at 468 nm with a quantum yield of 70%. By contrast, it presented a slight shift to 461 nm in the self-assembled state ($\Phi_Y$: 69%). The *R*-PEB also possessed strong blue fluorescence in both monomeric and aggregated states (Supplementary Fig. 12). In addition, the *S*-PEB supramolecular gel sample formed in MCH solution (11 mg/mL) had a maximum fluorescence peak at 498 nm and a shoulder peak at 526 nm, which showed a yellow-green fluorescence with a quantum yield of 39%. The *R*-PEB gel showed a similar yellow-green fluorescence (Supplementary Fig. 13). Moreover, their drop-casting thin film and bulk film (coassembly with poly(methyl methacrylate) (PMMA)) also showed strong fluorescence properties (Supplementary Figs. 14, 15). Detailed photophysical properties of *S*-PEB and *R*-PEB in various self-assembled states were summarized in Supplementary Table 1 and Supplementary Figs. 16–20.

Compared with the tedious preparation of multi-component CPL materials, single-component supramolecular materials with CPL-active property offer great advantages but remain challenging. Since the

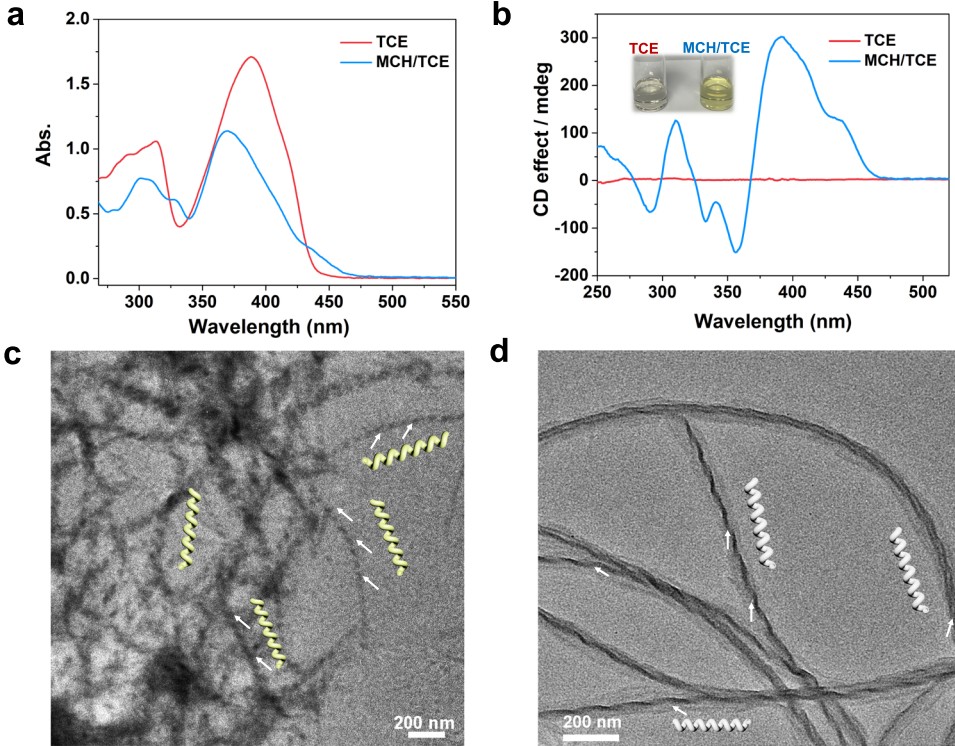

**Fig. 2 | UV-vis and CD spectra as well as TEM images. a** UV-vis absorption spectra of *S*-PEB in TCE and MCH/TCE (24:1, v/v) at 288 K. Source data are provided as a Source Data file. **b** CD spectra of *S*-PEB in TCE and MCH/TCE (24:1, v/v) at 298 K. The inset is the photograph of *S*-PEB in different states. Source data are provided as a Source Data file. TEM images of the supramolecular polymers of (**c**) *S*-PEB and (**d**) *R*-PEB prepared from MCH/TCE solution.

supramolecular solution of *S*-PEB and *R*-PEB formed ordered helical aggregated structures with intense fluorescence emission, hence we further investigated their circularly polarized luminescence (CPL) properties. As depicted in Fig. 4a, a strong negative CPL signal was located at 441 nm for *S*-PEB in MCH/TCE solution (24:1, v/v), and a mirror positive CPL signal was observed for *R*-PEB at the same condition. By contrast, *S*-PEB and *R*-PEB showed the absence of CPL signal in their TCE solution, manifesting that supramolecular polymerization of *S*-PEB and *R*-PEB contributed to the emergent CPL signals. The luminescence dissymmetry factor ($g_{lum}$) of *S*-PEB and *R*-PEB was calculated to be −0.011 and +0.011, respectively, which are much higher than that of many documented organic compounds in dilute solution[1]. It is worth noting that *S*-PEB and *R*-PEB are considered as ideal CPL active materials which integrate large luminescence factor ($10^{-2}$) and FL quantum efficiency (69%) in one molecule, which is not only the highest figure of merit (F = $g_{lum} \times \Phi_Y$ = 0.0076) in the single-component blue-color system but regards as a remarkable value among the reported supramolecular CPL materials so far. We then turned to explore the CPL properties of *S*-PEB and *R*-PEB gel samples. As shown in Fig. 4b, the *S*-PEB gel was detected a positive CPL signal located at 510−520 nm. The $g_{lum}$ and $\Phi_Y$ of the yellow-green CPL were determined as $5.3 \times 10^{-3}$ and 38%. To our surprise, from supramolecular solution to supramolecular gel, the CPL color of *S*-PEB is not only tuned from blue to yellow-green light but showed a handedness inversion of CPL signal with the same excitation wavelength. The *R*-PEB also presented this amazing reversal CPL property for corresponding supramolecular solution and gel samples. Unlike the reported work that the CPL inversion brought from supramolecule chirality inversion (CD signals)[50–52], it did not show chirality inversion between the supramolecular solution and gel sample from their CD spectra (Supplementary Fig. 21). It should be highlighted that they are rare examples that possess multi-color and multi-modal CPL behaviors with chirality unchanged[53].

In order to further verify this unique phenomenon, we obtained various CPL spectra by switching the excitation wavelength (Supplementary Fig. 22). All of them showed a strong negative CPL band ranging from 410 to 503 nm, accompanied by a weak positive emission band located from 504 to 610 nm. This phenomenon that one compound with two different handedness of CPL signals has been reported in previous literature[54–56]. To better understand the intrinsic cause, we applied theoretical calculations to simulate the CPL spectra in various states. The calculated CPL spectra (ECD) of one and two *S*-PEB molecules (dimer) were obtained by the method of B3LYP/6-31 G* (with methylcyclohexane as solvent), and the two cases both showed two different handedness of CPL signals (Supplementary Fig. 23). In addition, the CPL spectrum of *S*-PEB dimer presented an invented handedness ranging from 380 to 600 nm in comparison to the corresponding single molecule. These results indicate that the CPL signal of *S*-PEB is susceptible to the chromophore stacking layers, which is in good agreement with the inverted CPL presented by supramolecular solution and gel. Therefore, this color-dependent CPL property having a negative blue-color CPL signal and a positive yellow-green-color CPL signal should be derived from the intrinsic property of *S*-PEB.

Next, we prepared film samples of *S*-PEB and *R*-PEB. As shown in Fig. 5a, the drop-casting film of *S*-PEB presented positive yellow-green CPL, and an opposite CPL profile was observed for the *R*-PEB thin film. It is found that the CPL handedness of their thin films is both consistent with corresponding gels, but opposite with their supramolecular solution. In addition, the *S*-PEB and *R*-PEB coassembled with PMMA matric to obtain a homogeneous bulk film (*S*-PEB@PMMA and *R*-PEB@PMMA, detailed preparation is shown in the Supplementary Information). As displayed in Fig. 5b, S-PEB@PMMA possess yellow-green color CPL signal at around 510 nm, giving rise to a $g_{lum}$ of $10^{-3}$ and a $\Phi_Y$ of 26%. An identical CPL handedness from gel samples and drop-casting thin film of *S*-PEB was shown, which ensured the

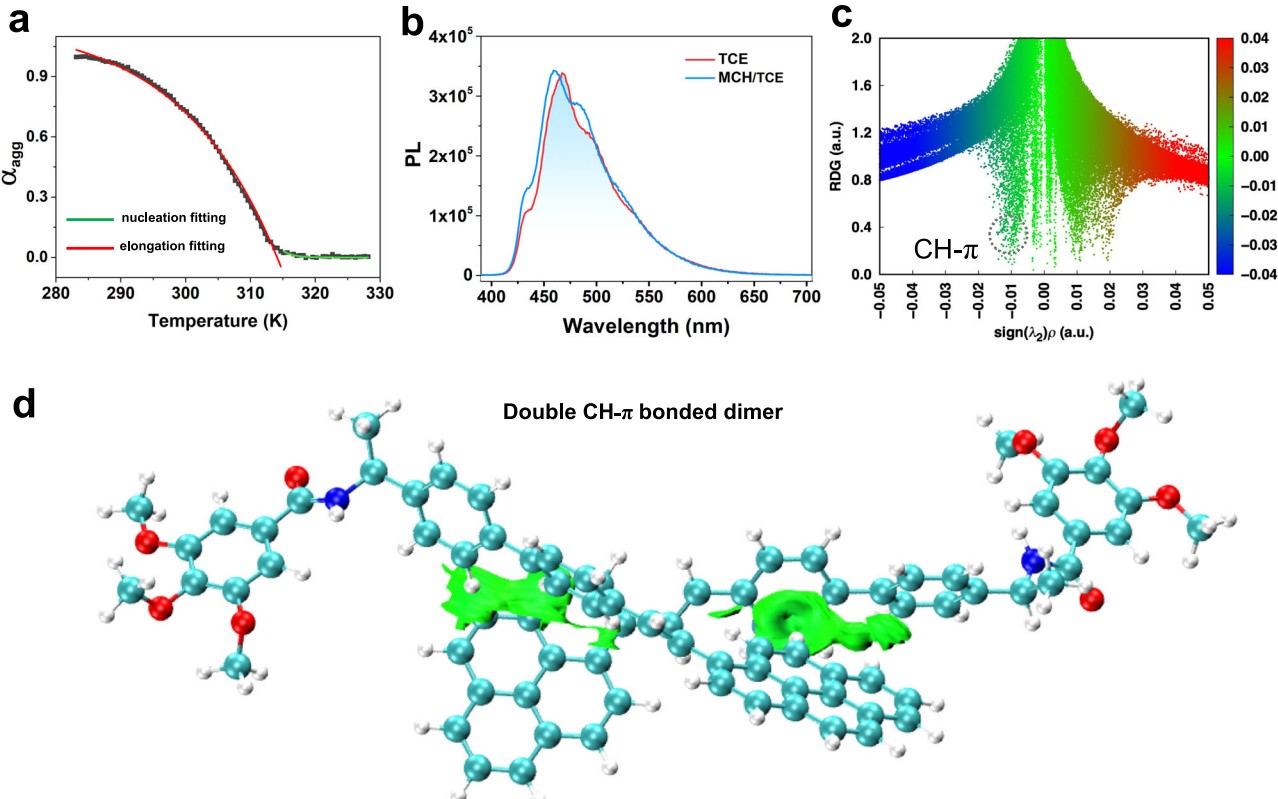

**Fig. 3 | CD intensity, fluorescence spectra, calculated scatter maps and iso-surface visualization. a** Normalized CD intensity of *S*-PEB at 387 nm versus temperature in MCH/TCE (24:1, v/v) at a heating rate of 1 K min$^{-1}$ ($C = 2 \times 10^{-4}$ M). The solid lines denote nucleation curve fitting (green line) and elongation curve fitting (red line), respectively. Source data are provided as a Source Data file. **b** Fluorescence spectra of *S*-PEB in dilute TCE and MCH/TCE (24:1, v/v) solution ($C = 4 \times 10^{-4}$ M). Source data are provided as a Source Data file. **c** Calculated scatter maps of the reduced density gradient (RDG) for *S*-PEB dimers relative to the space function sign ($\lambda_2$) ρ and visualization of the isosurface for weak intermolecular forces. **d** Visualization of the isosurface for existing multiple weak CH (phenyl group) · π (center of pyrene) contacts by Multiwfn and VMD. The green parts in the isosurface plots denote the multiple CH-π interactions.

authenticity of the positive signal of yellow-green color CPL. *R*-PEB presented similar CPL properties. It should be noted that four assembled samples of *S*-PEB and *R*-PEB all possess high-performance CPL properties (Supplementary Table 2). Moreover, the luminescence color and handedness of CPL are simultaneously tunable via varying supramolecular polymerization patterns, which is a rare case in the CPL field.

Styrylpyrene is a conventional photo-responsive unit, whose photoisomerization has been studied in dilute solution and crystal. Compared with monomeric solution and crystal samples, photo-isomeric supramolecular solution of styrylpyrene has not been well investigated so far. Take *S*-PEB for example (Fig. 6a), upon irradiating its MCH/TCE (24:1, v/v) solution (2 × 10$^{-4}$ M) with a 405 nm LED lamp (20 W), the absorption intensity at 389 nm gradually declined and two new peaks appeared at 331 nm and 348 nm, which suggested the occurrence of photo-induced [2 + 2] cycloaddition reaction of styrylpyrene[38,57,58]. The photo-stationary state (PSS) was observed upon irradiation for 80 s, indicating the coexistence of *S*-PEB and [2 + 2] cycloaddition products. The emerging NMR signals at 4.2, 5.1, and 5.8 ppm (for cyclobutane ring unit) and the absence signals at 6.5−6.7 ppm (for *cis*-isomer) collectively indicated that the dimer cycloaddition, rather than *trans-cis* isomerization, is the dominant reaction in supramolecular solution (Supplementary Fig. 24). Simultaneously, the CD signals of *S*-PEB supramolecular solution (4 × 10$^{-4}$ M) gradually decreased, illustrating the dissociation of chiral supramolecular aggregates (Supplementary Fig. 25). Irradiating the TCE solution of *S*-PEB (2 × 10$^{-4}$ M), similar spectrum was depicted in Fig. 6b, indicating the occurrence of the [2 + 2] cycloaddition reaction. Notably,

the PSS of *S*-PEB monomeric solution was determined upon irradiation for 480 s, which was much longer than its supramolecular one. The photoactivity rate constant ($K$) of monomeric and supramolecular solutions were determined to be 0.24 min$^{-1}$ and 2.53 min$^{-1}$ with the same condition, respectively (Fig. 6c, d). Notably, the $K_{MCH/TCE}$ (supramolecular state) is 10.5-fold larger than $K_{TCE}$ (monomeric state).

In addition, we also performed the photoreaction in THF and MCH/THF systems (Supplementary Fig. 26). We found that the photo-response activity of *S*-PEB in dilute THF solution is lower than its mixed MCH/THF (24:1, v/v) solution, and the $K_{MCH/THF}$ is 4.4 times higher than $K_{THF}$. Therefore, it can be concluded that the supramolecular interactions make a major contribution to accelerating the light-induced [2 + 2] cycloaddition, which has not been reported in previous styrylpyrene systems. More interestingly, we found that neither the gel sample nor the self-assembled solid of *S*-PEB did not undergo the [2 + 2] cycloaddition under the same condition. Based on previous studies[59,60], there are three factors that are vital for the occurrence of the [2 + 2] cycloaddition reaction. (I) Distance criteria: the center-to-center distance of C = C groups suitable for cycloaddition is 3.7−4.2 Å. (II) Parallelism of double bonds. (III) Minimum translational movement. Regarding the supramolecular gel and self-assembled solid, the adjacent *S*-PEB molecules are connected to form tight networks via supramolecular interactions. In this case, although it meets distance criteria, the rotation angle of adjacent C = C units may greatly increase due to overcrowded stacking, which may inhibit the process of cycloaddition[61,62]. For gel and powder samples, their tight stacking patterns possibly restrict the spatial movement of the C = C skeleton, preventing them from obtaining ideal geometries that are conducive

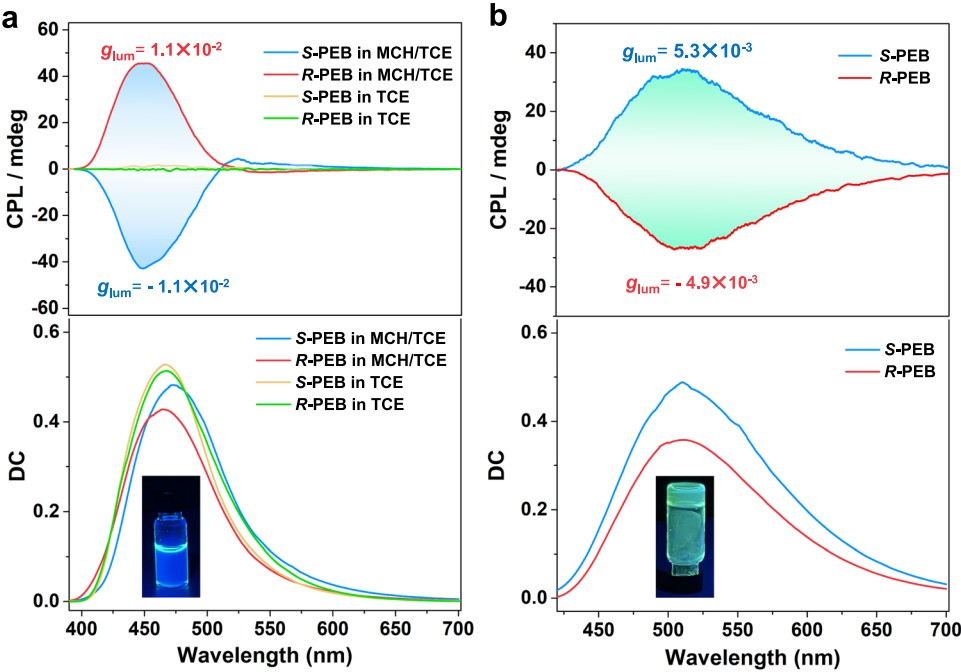

**Fig. 4 | CPL spectra. a** CPL spectra of *S*-PEB and *R*-PEB in MCH/TCE and TCE solution ($C = 4 \times 10^{-4}$ M). Inset: the fluorescence image of *S*-PEB in MCH/TCE solution. **b** CPL spectra of *S*-PEB and *R*-PEB gel samples formed in MCH solution (11 mg/mL). Inset: the fluorescence image of *S*-PEB gel. Source data are provided as a Source Data file.

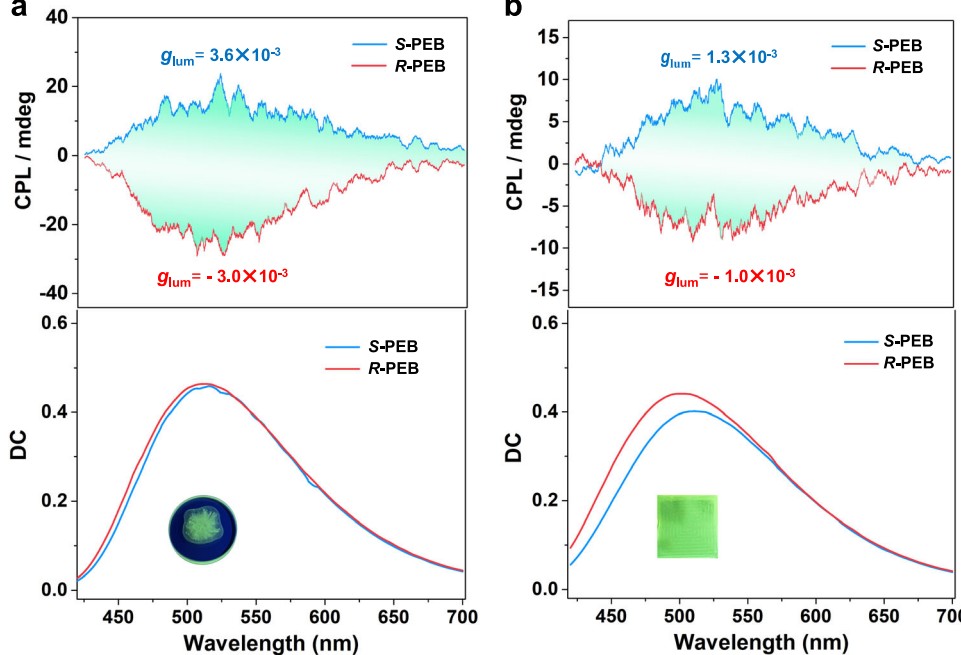

**Fig. 5 | CPL spectra. a** CPL spectra of drop-casting thin films of *S*-PEB and *R*-PEB prepared from MCH/TCE solution ($C = 4 \times 10^{-4}$ M). Inset: the fluorescence image of *S*-PEB thin film. **b** CPL spectra of *S*-PEB and *R*-PEB @PMMA films. Inset: the fluorescence image of *S*-PEB@PMMA bulk film. All data are the average of three parallel tests. Source data are provided as a Source Data file.

to [2 + 2] cycloaddition reactions. Therefore, the gel and supramolecular solid are basically photostable under light exposure. Furthermore, the supramolecular solution is a dynamic equilibrium between monomeric and supramolecular states, which combines the mobility and aggregation of molecules. Therefore, the supramolecular confinement effect promoted higher local concentrations of *S*-PEB, leading to accelerated [2 + 2] photo-cycloaddition reactivity in the current

system. Overall, by precisely controlling the photochemical reactivity through supramolecular polymerization, the present study opens up new thoughts toward cycloaddition reaction with high synthetic efficiency.

To clarify the structure of photo-induced [2 + 2] cycloaddition products, three parallel irradiation tests (10 mL, 0.0005 M) were performed in TCE and MCH/TCE (24:1, v/v) solution, respectively. After

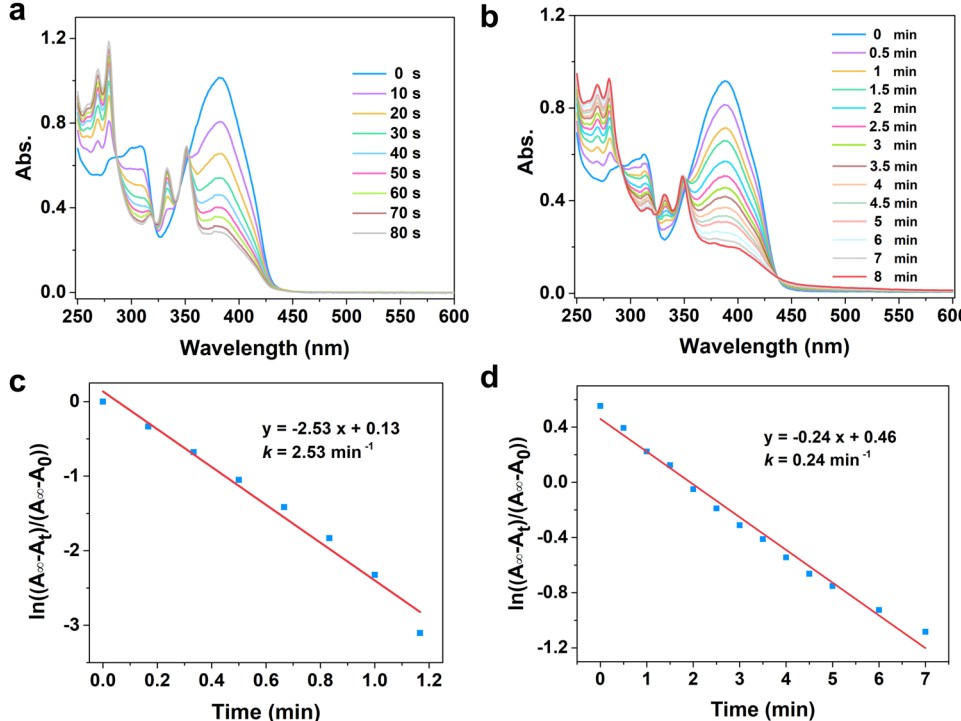

**Fig. 6 | Time-dependent UV-vis spectra and kinetics.** Time-dependent UV-vis spectra of *S*-PEB in (**a**) MCH/TCE (24:1, v/v) and (**b**) TCE solution upon 405 nm light source irradiation. Kinetic studies for the [2 + 2] cycloaddition reaction of *S*-PEB in (**c**) MCH/TCE and (**d**) TCE solution at the same condition ($C = 2 \times 10^{-4}$ M, optical path length: 1 mm). All reaction solutions are purged with a stream of argon for 30 min before photoreaction. Source data are provided as a Source Data file.

purification, we obtained two cycloaddition products with different configurations (Dm-PEB-1 and Dm-PEB-2). Both of them are characterized by $^1$H NMR and 2D NOESY (Supplementary Figs. 27–31). The photophysical and chiroptical properties were shown in Supplementary Fig. 32. No CD and CPL feature was detected for both Dm-PEB-1 and Dm-PEB-2 in MCH solution, manifesting the lack of ordered chiral aggregates. For the TCE solution, after 210 min irradiation, apart from the remaining 55% of *S*-PEB, 7% of Dm-PEB-1, 6% of Dm-PEB-2, and an unknown by-product were obtained. For the supramolecular system, the content of *S*-PEB, Dm-PEB-1, and Dm-PEB-2 are 25%, 11%, and 46% upon irradiation for 55 min, respectively. No *cis*-isomer of *S*-PEB is obtained for both supramolecular solution and monomeric solution. Thus, the supramolecular solution is indeed more efficient to undergo [2 + 2] cycloaddition. More importantly, the yield of Dm-PEB-2 is about 4-fold higher than Dm-PEB-1, which is a favorable proof that *S*-PEB predominantly adopts the twist-antiparallel arrangement in supramolecular assembled states (Fig. 3d).

## Discussion

In summary, we have successfully constructed a single-component CPL material based on styrylpyrene. With the help of CH-π interactions, the styrylpyrene-cored molecule enables to form the long-range ordered helical fiber. Benefiting from the absence of π-π stacking, the $g_{lum}$ and $\Phi_Y$ of their supramolecular solution are determined to be $1.1 \times 10^{-2}$ and 69%, respectively. In addition, the supramolecular gel and film samples also show excellent CPL properties ($g_{lum} = 1 \times 10^{-3} - 5.3 \times 10^{-3}$ and $\Phi_Y = 28\% - 38\%$). Intriguingly, from the supramolecular solution to supramolecular gel or thin film, not only the emission color is shifted from blue to yellow-green, but also the handedness of CPL is inverted while the supramolecular chirality remains unchanged, thus marking it a unique multi-modal and color-dependent single-component CPL-active material. Moreover, with the aid of the supramolecular confinement effect, the reaction rate of photo-induced [2 + 2] cycloaddition for supramolecular solution is 10.5 times higher than its

monomeric solution. In contrast, no cycloaddition reaction occurs for the gel and self-assembled solid, which is a rare finding in the field of cycloaddition reaction. This strategy, in turn, may aid the future design of versatile CPL-active materials with facile modulation in the color and handedness of CPL, and facilitate to construct chiral photo-responsive devices.

## Methods
### Materials
Unless otherwise noted, all commercial chemicals were used as received without further purification. Palladium(II) acetate (purity: 99%) and XPhos ligand (purity: 98%) were purchased from Aldrich. Bis(pinacolato) diboron (purity: 98%), 1-bromopyrene (purity: 95%), Pd(PPh$_3$)$_4$ (purity: 98%), Pd$_2$(dba)$_3$·CHCl$_3$ (purity: 98%), (*S*) and (*R*)-1-(4-bromophenyl)ethan-1-amine (purity: 98%) were purchased from the BLD Pharmatech Ltd. EDC·HCl (purity: 98%) and DMAP (purity: 98%) were purchased from the TCI. Dichloromethane (CH$_2$Cl$_2$) was distilled over CaH$_2$. All reactions were performed under an atmosphere of argon and monitored by TLC with silica gel 60 F254. Column chromatography was carried out on silica gel (230–400 mesh).

### Characterization
$^1$H NMR (400 MHz) and $^{13}$C NMR (100 MHz) spectra were recorded on a Bruker BBFO 400 Spectrometer. Electrospray ionization (ESI) mass spectra were recorded on a ThermoFinnigan LCQ quadrupole ion trap mass spectrometer. Absorption spectra were collected on UV-3600 Shimadzu Spectrophotometer. Fluorescence spectra were recorded on a Horiba Fluoromax-4 spectrofluorometer. Fluorescence quantum yields were measured using a Hamamatsu absolute photoluminescent (PL) quantum yield spectrometer. Transient PL decay spectra were measured under nitrogen atmosphere (solution) or vacuum (neat film), using the Quantaurus-Tau fluorescence lifetime measurement system (C11367-03, Hamamatsu Photonics Co., Japan). Circular dichroism (CD) measurements were performed on a Jasco J-1500

circular dichroism spectrometer, equipped with a PFD-425S/15 Peltier-type. Circularly polarized luminescence spectra were recorded on a CPL-300 spectrometer. TEM images were carried on an EOL JEM 1400 microscope. Synthetic scheme is shown in Supplementary Fig. 1 and characterization spectra are presented in Supplementary Figs. 33–38.

## General synthetic procedure for compound B₁

Palladium(II) acetate (56 mg, 0.25 mmol) and XPhos ligand (0.24 g, 0.5 mmol) were placed under an argon atmosphere in a Schlenk tube and DMF (20 mL) was added. After stirring for 15 min, 1-bromopyrene (1.40 g, 5 mmol), 1-chloro-4-vinylbenzene (0.69 g, 5 mmol), and $K_2CO_3$ (1.38 g, 10 mmol) were added. Subsequently, the mixture was heated at 60 °C for 10 h. After adding water and $CH_2Cl_2$ (100 mL), the organic and the aqueous layers were separated. The combined organic layers were dried by $Na_2SO_4$. After filtration, the filtrate was concentrated in vacuo. The residue was purified by a silica column chromatography (the eluent: hexane/ethyl acetate) to obtain B₁ as a yellow solid. ¹H NMR (400 MHz, CDCl₃) δ 8.48–8.46 (d, 1H), 8.31–8.29 (d, 1H), 8.20–8.13 (m, 5H), 8.06–7.99 (m, 3H), 7.62–7.60 (d, 2H), 7.41–7.39 (d, 2H), 7.31–7.27 (d, 1H). These data are consistent with the literature report[36].

## General synthetic procedure for compound S-A

Compound A₁ (1.80 g, 2.66 mmol), (S)-1-(4-bromophenyl)ethan-1-amine (0.41 g, 2.22 mmol), DMAP (0.45 g, 3.73 mmol) and EDC·HCl (1.03 g, 5.35 mmol) were mixed in $CH_2Cl_2$ (50 mL) under argon and stirred at room temperature for 12 h. The solution was extracted with $H_2O/CH_2Cl_2$ for three times. The solvent was evaporated with a rotary evaporator. The residue was purified by silica column chromatography (the eluent: hexane/$CH_2Cl_2$) to provide S-A as a white solid (1.48 g, 70%). ¹H NMR (400 MHz, CDCl₃) δ 7.48–7.46 (d, 2H), 7.27–7.25 (d, 2H), 6.94 (s, 2H), 6.18–6.16 (s, 1H), 5.26–5.23 (q, 1H), 3.99–3.98 (m, 6H), 1.81–1.71 (m, 6H), 1.56 (d, 3H), 1.46–1.26 (m, 60H), 0.90–0.86 (m, 9H). These data are consistent with the literature report[18].

## General synthetic procedure for compound R-A

Compound A₁ (2.00 g, 2.96 mmol), (R)-1-(4-bromophenyl)ethan-1-amine (0.46 g, 2.47 mmol), DMAP (0.50 g, 4.15 mmol) and EDC·HCl (1.14 g, 5.95 mmol) were mixed in $CH_2Cl_2$ (50 mL) under argon and stirred at room temperature for 12 h. The solution was extracted with $H_2O/CH_2Cl_2$ for three times. The solvent was evaporated with a rotary evaporator. The residue was purified by silica column chromatography (the eluent: hexane/$CH_2Cl_2$) to provide R-A as a white solid (1.48 g, 70%). ¹H NMR (400 MHz, CDCl₃) δ 7.48–7.46 (d, 2H), 7.27–7.25 (d, 2H), 6.93 (s, 2H), 6.17–6.15 (s, 1H), 5.26–5.23 (q, 1H), 3.99–3.98 (m, 6H), 1.81–1.71 (m, 6H), 1.56 (d, 3H), 1.46–1.26 (m, 60H), 0.90–0.86 (m, 9H). These data are consistent with the literature[18].

## General synthetic procedure for compound B₂

A mixture of B₁ (0.80 g, 2.37 mmol), bis(pinacolato)diboron (0.90 g, 3.55 mmol), NaOAc (0.58 g, 7.11 mmol), Pd₂(dba)₃·CHCl₃ (0.19 g, 0.19 mmol) and XPhos (0.19 g, 0.38 mmol) in dry 1,4-dioxane (50 mL) was stirred at 110 °C under argon atmosphere for 24 h. After cooling to room temperature, the mixture was poured into $H_2O$ (150 mL) and extracted with dichloromethane (200 mL). The combined organic layer was washed with water and dried by $MgSO_4$. The solvent was removed in vacuo and the residue was purified by column chromatography (the eluent: hexane/ $CH_2Cl_2$) on silica gel to get yellow solid B₂ (0.35 g, 0.81 mmol, 34%). ¹H NMR (400 MHz, CDCl₃ ppm) δ 8.51–8.48 (d, 1H), 8.33–8.24 (m, 2H), 8.19–8.12 (m, 4H), 8.05–7.98 (m, 3H), 7.91–7.89 (d, 2H), 7.71–7.69 (d, 2H), 7.38–7.34 (d, 1H), 1.40 (s, 12H). ¹³C NMR (100 MHz, CDCl₃, ppm): δ 140.5, 135.4, 131.9, 131.7, 131.1, 128.6, 127.8, 127.6, 127.5, 126.8, 126.1, 125.5, 125.3, 125.2, 125.1, 123.8, 123.2, 84.0, 25.1. HR-MS, m/z: calculated, 430.2104; found, 431.2180 ([M + H]⁺).

## General synthetic procedure for S-PEB

A mixture of S-A (0.30 g, 0.35 mmol), B₂ (0.15 g, 0.35 mmol), NaHCO₃ (0.19 g, 1.40 mmol) and Pd(PPh₃)₄ (0.04 g, 0.035 mmol) in dioxane/ $H_2O$ (10 mL/2 mL) was stirred at 110 °C under argon atmosphere for 24 h. After cooling to room temperature, the mixture was poured into $H_2O$ (15 mL) and extracted with dichloromethane (40 mL). The combined organic layer was washed by water and dried with $MgSO_4$. The solvent was removed in vacuo and the residue was purified by column chromatography (the eluent: hexane/ethyl acetate) on silica gel to get yellow solid S-PEB (0.20 g, 0.19 mmol, 53%). ¹H NMR (400 MHz, CDCl₃, ppm): δ 8.53–8.51 (d, 1H), 8.36–8.34 (d, 1H), 8.27–8.01 (m, 8H), 7.78–7.76 (d, 2H), 7.68–7.66 (d, 4H), 7.51–7.49 (d, 1H), 7.42–7.38 (d, 1H), 6.99 (s, 2H), 6.26–6.24 (d, 1H), 5.40–5.36 (m, 1H), 4.03–3.99 (m, 6H), 1.83–1.66 (m, 10H), 1.5–1.26 (m, 51H), 0.89–0.86 (m, 9H). ¹³C NMR (100 MHz, CDCl₃, ppm): δ 166.7, 153.4, 142.6, 141.7, 140.2, 140.1, 137.1, 132.1, 131.8, 131.5, 131.1, 129.7, 128.7, 127.9, 127.7, 127.6, 127.5, 127.5, 127.4, 127.0, 126.2, 126.1, 125.5, 125.4, 125.3, 123.9, 123.2, 106.2, 73.7, 69.8, 49.2, 32.1, 29.9, 29.9, 29.8, 29.8, 29.6, 29.6, 29.6, 26.3, 22.9, 14.3. HR-MS, m/z: calculated, 1079.7731; found, 1080.7872 ([M + H]⁺).

## General synthetic procedure for R-PEB

The synthesis of R-PEB was performed analogously to that of S-PEB. The product was obtained as a yellow solid (46%). ¹H NMR (400 MHz, CDCl₃, ppm): δ 8.52 (d, 1H), 8.36–8.01 (m, 9H), 7.78–7.67 (d, 6H), 7.50–7.38 (m, 3H), 6.99 (s, 2H), 6.26–6.24 (d, 1H), 5.40–5.36 (m, 1H), 4.03–3.99 (m, 6H), 1.83–1.66 (m, 9H), 1.5–1.26 (m, 53H), 0.89–0.86 (m, 10H). ¹³C NMR (100 MHz, CDCl₃, ppm): δ 166.7, 153.4, 142.6, 137.1, 131.5, 127.87, 127.7, 127.6, 127.5, 127.4, 127.0, 126.2, 126.1, 125.6, 125.4, 125.3, 123.9, 106.2, 73.7, 69.8, 49.2, 32.1, 30.5, 29.9, 29.8, 29.8, 29.6, 29.6, 26.3, 22.9, 21.9, 14.3. HR-MS, m/z: calculated, 1079.7731; found, 1080.7849 ([M + H]⁺).

## Preparation of S-PEB@PMMA and R-PEB@PMMA coassembly films for CPL measurements

PMMA (80 mg, Mn = 3,50,000) was dissolved in THF (3 mL). This process required stirring for 6 h and named it A solution. S-PEB (2.4 mg) was dissolved in THF (400 μL), which was named B solution. The solutions A (375 μL) and B (34 μL) were mixed and then dropped into a glass cuvette (20 mm × 20 mm × 0.5 mm). The uniform coassembly film of S-PEB@PMMA was generated in the cell after the slow evaporation of the solvent. The weight ratio of S-PEB: PMMA is 2:100. The preparation of R-PEB@PMMA film was the same as that of S-PEB@PMMA.

## Photo-induced [2 + 2] cycloaddition of S-PEB in monomeric and supramolecular solution for kinetic studies by UV-vis spectra

All solutions ($C = 2 \times 10^{-4}$ M, V = 350 μL) are placed in a cuvette with a screw cap (optical path length: 1 mm) and purged with a stream of argon for 30 min before irradiation by a 405 nm LED lamp. The lamp and sample are placed in a black box during the irradiation. The distance between the cuvette and the lamp is 15 cm.

## Photo-induced [2 + 2] cycloaddition of S-PEB in monomeric and supramolecular solution for purification of cycloaddition products

All solutions ($C = 5 \times 10^{-4}$ M, V = 10 mL) were placed in a round-bottom flask and purged with a stream of argon for 60 min before irradiation by a 405 nm LED lamp. TCE (three parallel tests) and MCH/TCE (24:1, v/v) solution (three parallel tests) were performed, respectively. UV absorption spectroscopy was used to detect the progress of the photoreaction. After reaching their PSS, merge the same type of solution and evaporate the solvent to dryness. Two photoproducts with different configurations (Dm-PEB-1 and Dm-PEB-2) were carefully separated via preparative thin-layer chromatography (TLC). HR-MS of

Dm-PEB-1 and Dm-PEB-2 are 2160.5715 ([M + H]$^+$) and 2160.5540 ([M + H]$^+$), respectively.

## Simulation details

The structures of helical stacking dimers were fully optimized by using the density functional theory (DFT) at the B3LYP theory of level[63,64]. The 6-31 g(d) basis set was adopted for geometry optimization. More explicitly, the geometry optimization was first performed for the single S-PEB, and the optimized S-PEB structure was used for constructing the dimeric S-PEB with the "head-to-tail" arrangement by matching the vinyl groups closely (that is where the polymerization happens). Then, the geometry optimization was performed on such dimeric S-PEB. Frequency calculations under the same theory of level were performed to confirm that the optimized structures were minima on the potential energy surface and no imaginary frequency was observed. All calculations were performed using Gaussian 16 package[65]. To capture weak interactions between stacking dimers, the reduced density gradient (RDG) method and the independent gradient model based on Hirshfeld partition (IGMH) were carried for analyzing the possible π-π stacking, hydrogen bond, and the CH·π interactions. The simulated circularly polarized luminescence spectra (ECD) were obtained by the TD-DFT calculation at the B3LYP/6-31 G* level. The molecular structures (for both single monomer and dimer) for the ground state (S$_0$) were used as the initial guesses for the optimizations of S$_1$ state (singlet excited state), and 10 states were used. All these post-processing analyses were performed using Multiwfn package (version 3.8 (dev), release date: 2022-Sep-12)[48].

## Data availability

The authors declare that all the data supporting the findings of this study are available within the article. The Supplementary Information, Source Data file, and full image dataset are available from the corresponding author upon request. Source data are provided with this paper.

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

## Acknowledgements

This work was supported by the Singapore Agency for Science, Tech-nology and Research (A*STAR) under Its Manufacturing, Trade and Connectivity Individual Research Grant (M22K2c0077, Y.Z.).

## Author contributions

W.Y. and Y.Z. conceived the ideas and conducted the experiments. L.C. helped with CPL experiments. C.Y. helped with the mechanism of supramolecular polymerization. Z. Zhang helped with the DFT analysis. X.C. and X.Z. performed TEM experiments. J.G. and C.Q. helped with the manuscript revision. W.Y. and Y.Z. co-wrote the manuscript. Z. Zhao and Y.Z. discussed the results and revised the manuscript. All authors have given approval to the final version of the manuscript.

## Competing interests

The authors declare no competing interests.
