## [Peer review file · Nature Communications]

REVIEWER COMMENTS

Reviewer #1 (Remarks to the Author):

The manuscript reported a new strategy to construct high-performance CPL materials. The varied aggregated states such as supramolecular solution, gel, thin film and co-assembled PMMA bulk film all showed high fluorescence quantum yields and g_{lum} , which were ascribed to the suppression of n-n stacking interaction. In addition, the photo-triggered cycloaddition reaction rate of aggregated solution was higher than its monomeric solution, and the cycloaddition products were controlled by the supramolecular interactions. Overall, this work is systematic and rather interesting for the photochemistry field. The paper could be accepted for publication following the minor revisions detailed below.

1. The formation of molecular aggregates for S-PEB was verified by the UV-vis and CD spectra. Apart from these experiments, the DLS experiments for the supramolecular solution (MCH/TCE) should be added to confirm the size of aggregates.
2. Regarding the driving force for the supramolecular polymerization, the authors highlighted the absence of n-n stacking interaction as certified by the concentration-dependent NMR, fluorescence spectra and DFT calculations. In addition to these studies, concentration-dependent UV-vis experiments of S-PEB in the mixture solvent (MCH/TCE) should be performed.
3. For the [2+2] photoreaction part (Figure 5), the authors should give more detailed information about the light source such as the distance between the lamp and the sample and the power of the LED lamp used.

Reviewer #2 (Remarks to the Author):

In this manuscript, Zhao et al. present their findings on high-performance CPL materials achieved through the incorporation of chirality units into a styrylpyrene core for supramolecular polymerization. The resulting materials demonstrated a significant luminescence dissymmetry factor ($g_{lum} = 1.1 \times 10^{-2}$) and a high fluorescence quantum yield ($\Phi_Y = 69\%$) upon forming self-assembled supramolecular solutions. The authors propose that the main driving force behind the construction of chiral supramolecular structures is the CH-n interaction almost only by theoretical calculation. Furthermore, the supramolecular solutions exhibited a faster photocycloaddition reaction rate compared to the monomeric solutions. Based on these findings, I think this manuscript may be accepted with major revisions.

1. Indeed, the excellent CPL performance achieved in this study addresses the inherent contradiction between the efficiency of CPL materials and their luminescence dissymmetry factor. It would be beneficial to summarize the CPL performance of the self-assembled materials in a table to highlight their advantages. By presenting the CPL performance of the self-assembled materials in a table, the advantages of these materials become more apparent and can be easily compared to the reported materials.
2. The evidences for confirming the formation of chiral supramolecular structures through CH-n interaction is important in this study. The characterization of CH-n interaction seems to be inefficient. XPS would be helpful. What's the dimer for RDG calculation achieved? From the single crystal structure or just by theoretical optimization? This should be clarified and justified.
3. In some cases, the full term of the abbreviation is not given- examples: "MCH". Please check carefully.
4. Please provide a photograph of the supramolecular solution in MCH/TCE. Is it clear and transparent, or is it turbid? Does it exhibit the Tyndall effect?

5. CPL materials (R/S-PEB) are formed by combining chirality units with a styrylpyrene core. However, the manuscript discusses chirality and photocycloaddition in separated aspects without integrating them. What is the chiral performance of the cycloaddition products?

6. As mentioned by the authors, styrylpyrene is a conventional photo-responsive unit. However, compared to monomeric solutions and crystals, supramolecular solutions exhibit enhanced photocycloaddition reactivity. The authors did not provide a clear explanation for this observation in this part of the manuscript.

7. Since this study is mainly on photoluminescence. I suggest the authors demonstrate at least one set of PL spectra in the main text, NOT only CPL spectra.

8. The reasons for the inverted CPL are too simple and need to be discussed in detail. Will it be related to the different CH- π interaction modes in different states?

Reviewer #3 (Remarks to the Author):

Developing facile and efficient methods to obtain circularly polarized luminescence (CPL) materials with a large luminescence dissymmetry factor (g_{lum}) and fluorescence quantum yield (Φ_Y) is attractive but still challenging. In this nice manuscript, supramolecular polymerization of styrylpyrenes (R/S-PEB) is utilized to attain this aim, which can self-assemble into helical nanoribbons driven by unusual CH- π interactions. Supramolecular solution of S-PEB shows remarkable blue-color CPL property (g_{lum} : 0.011, Φ_Y : 69%). From supramolecular solution to gel, the emission color (blue to yellow-green) and handedness of CPL (g_{lum} : - 0.011 to + 0.005) are concurrently manipulated, while the corresponding supramolecular chirality maintains unchanged, representing the rare example of color-dependent CPL materials. Benefiting from supramolecular confinement effect, the [2+2] cycloaddition reaction rate of the supramolecular solution is 10.5 times higher than that of the monomeric solution. In contrast, no cycloaddition reaction occurs for the gel and assembled solid samples. Our findings provide a new vision for fabricating multi-modal and high-performance CPL-active materials, paving the way for the development of advanced photo-responsive chiral devices. The work presented in this manuscript has good novelty and scientific value. The text is well organized and the results are clearly discussed. Therefore, I recommend this manuscript to be accepted after finishing the following minor revisions:

1. In the abstract, change "that of monomeric solution" to "that of the monomeric solution".
2. In the introduction, change "their promising application in 3D display" to "their promising applications in 3D display".
3. In the introduction, change "multi-color property are rarely reported" to "multi-color property have been rarely reported".

Response to Reviewers' Comments

Responses to Reviewer 1:

General comments: *The manuscript reported a new strategy to construct high-performance CPL materials. The varied aggregated states such as supramolecular solution, gel, thin film and co-assembled PMMA bulk film all showed high fluorescence quantum yields and glum, which were ascribed to the suppression of π - π stacking interaction. In addition, the photo-triggered cycloaddition reaction rate of aggregated solution was higher than its monomeric solution, and the cycloaddition products were controlled by the supramolecular interactions. Overall, this work is systematic and rather interesting for the photochemistry field. The paper could be accepted for publication following the minor revisions detailed below.*

Author response: we thank the reviewer's positive comments on our work and recommendation for publication after minor revisions.

1. The formation of molecular aggregates for S-PEB was verified by the UV-vis and CD spectra. Apart from these experiments, the DLS experiments for the supramolecular solution (MCH/TCE) should be added to confirm the size of aggregates.

Author response: We thank the reviewer's kind suggestions. As suggested, the DLS measurement of the **S-PEB** solution (MCH/TCE, 4×10^{-4} M) has been performed and shown in Supplementary Fig. 2. The average size of **S-PEB** aggregates is 537.7 ± 186.3 nm (Please check Page 7 for the discussion).

2. Regarding the driving force for the supramolecular polymerization, the authors highlighted the absence of π - π stacking interaction as certified by the concentration-dependent NMR, fluorescence spectra and DFT calculations. In addition to these studies, concentration-dependent UV-vis experiments of S-PEB in the mixture solvent (MCH/TCE) should be performed.

Author response: We thank the reviewer's kind comments. We have performed the concentration-dependent UV-vis measurement of **S-PEB** solutions and these data have been added in Supplementary Fig. 1 (Please check Page 7 for the discussion).

3. For the [2+2] photoreaction part (Figure 5), the authors should give more detailed information about the light source such as the distance between the lamp and the sample and the power of the LED lamp used.

Author response: We appreciate the reviewer's kind reminder. This information has been added in the part of experimental procedure (Please check Page 4 in the Supplementary Information).

Responses to Reviewer 2:

General comments: *In this manuscript, Zhao et al. present their findings on high-performance CPL materials achieved through the incorporation of chirality units into a styrylpyrene core for supramolecular polymerization. The resulting materials demonstrated a significant luminescence dissymmetry factor ($g_{lum} = 1.1 \times 10^{-2}$) and a high fluorescence quantum yield ($\Phi Y = 69\%$) upon forming self-assembled supramolecular solutions. The authors propose that the main driving force behind the construction of chiral supramolecular structures is the CH- π interaction almost only by theoretical calculation. Furthermore, the supramolecular solutions exhibited a faster photocycloaddition reaction rate compared to the monomeric solutions. Based on these findings, I think this manuscript may be accepted with major revisions.*

Author response: We sincerely thank the reviewer for the useful comments and advice on our work. We have done our best to carry out additional experiments to support the conclusions and further improve the accuracy of the manuscript.

1. Indeed, the excellent CPL performance achieved in this study addresses the inherent contradiction between the efficiency of CPL materials and their luminescence dissymmetry factor. It would be beneficial to summarize the CPL performance of the self-assembled materials in a table to highlight their advantages. By presenting the CPL performance of the self-assembled materials in a table, the advantages of these materials become more apparent and can be easily compared to the reported material.

Author response: We sincerely thank the reviewer's kind suggestion. The summarized information has been updated in the Supplementary Table 2.

2. The evidences for confirming the formation of chiral supramolecular structures through CH- π interaction is important in this study. The characterization of CH- π interaction seems to be inefficient. XPS would be helpful. What's the dimer for RDG calculation achieved? From the single crystal structure or just by theoretical optimization? This should be clarified and justified.

Author response: We sincerely appreciate the reviewer for his/her useful comments and advice. As suggested, firstly, the ^1H - ^1H COSY and NOESY NMR experiments have been employed to reveal the interactions among the protons. As shown in Supplementary Fig. 8, we marked the position of all protons. From the 2D NOESY spectrum of **S-PEB** (Supplementary Fig. 9), we can find that the cross peak of proton **2'** and **11'** existed, indicating the twisted "head-to-tail" arrangement. These results are in good agreement with the DFT results (Fig. 2d). Apart from the 2D NMR characterization, we obtained the cycloaddition products of **S-PEB** supramolecular solution under irradiation (405 nm). Two main products are **Dm-PEB-1** and **Dm-PEB-2** (Supplementary Fig. 26), indicating the presence of two proposed stacking modes. The ratio of **Dm-PEB-1**: **Dm-PEB-2** is 0.25, suggesting that the twist-antiparallel (head-to-tail) is the dominant packing pattern. In addition, the concentration-dependent NMR experiments and fluorescence spectra of monomeric and aggregated states jointly exclude the existence of π - π interaction in the supramolecular states. Thus, based on the data

obtained above, it is reasonable to confirm the predominant presence of CH- π interactions between the **S-PEB** moieties in the supramolecular solution state (Please check Page 10 for the discussion).

Figure R1. (a) Survey XPS spectrum of the **S-PEB** powder assembled from supramolecular solution and (b) high-resolution XPS peak of C 1s.

Then, the XPS experiment of **S-PEB** powder self-assembled from MCH/TCE solution has been performed (Figure R1). Although the analyzed data can confirm the presence of C-C, C=C, and C-N bonds in the π -system, the intermolecular CH- π interaction cannot be completely verified by using this method.

Regarding the dimer for RDG, we have synthesized the model molecule **PEB-0**. We are unable to get high-quality crystals for X-ray diffraction experiments. Thus, based on the previous literature (*Eur. J. Org. Chem.* **2011**, 5261) and experimental data, the dimer for RDG was obtained from geometry optimization in the gas phase.

3. In some cases, the full term of the abbreviation is not given- examples: "MCH". Please check carefully.

Author response: Thanks! The full name of the abbreviation "MCH" is given (Please check Page 7).

4. Please provide a photograph of the supramolecular solution in MCH/TCE. Is it clear and transparent, or is it turbid? Does it exhibit the Tyndall effect?

Author response: We thank the reviewer's kind suggestions. We have added the picture of **S-PEB** in monomeric and supramolecular solution (Fig. 1b). The supramolecular solution is clear and shows a Tyndall effect (Supplementary Fig. 2).

5. CPL materials (R/S-PEB) are formed by combining chirality units with a styrylpyrene core. However, the manuscript discusses chirality and photocycloaddition in separated aspects without integrating them. What is the chiral performance of the cycloaddition products?

Author response: We thank the reviewer's kind comments and suggestions. We have synthesized and purified the cycloaddition products **Dm-PEB-1** and **Dm-PEB-2**. As shown in Supplementary Fig. 31, both **Dm-PEB-1** and **Dm-PEB-2** were not CPL-active material, due to the lack of long-range ordered arrangement (Please check Page 18 for the discussion).

6. As mentioned by the authors, styrylpyrene is a conventional photo-responsive unit. However, compared to monomeric solutions and crystals, supramolecular solutions exhibit enhanced photo-cycloaddition reactivity. The authors did not provide a clear explanation for this observation in this part of the manuscript.

Author response: We greatly thank the reviewer's important comments. We have performed the photoreaction of the gel and self-assembled powder under the same conditions, and found that no cycloaddition product was obtained, which is unambiguously confirmed through the silica thin layer chromatography (TLC) method and UV-vis spectroscopy. It is noteworthy that three factors are vital for the occurrence of the [2+2] cycloaddition reaction. (I) Distance criteria: the center-to-center distance of C=C groups suitable for cycloaddition (so far accepted limit is 3.7-4.2 Å). (II) Parallelism of double bonds: the ideal rotational angle of C=C groups is 0°. (III) Minimum translational movement (*Chem. Rev.* **1987**, *87*, 433; *J. Am. Chem. Soc.* **1973**, *95*, 2324). Regarding the supramolecular gel and self-assembled solid, the adjacent molecules are connected to form tight networks via supramolecular interactions. In this case, although it meets the distance criteria (*Angew. Chem. Int. Ed.* **2017**, *56*, 9463), the rotation angle of adjacent C=C units may greatly increase due to overcrowded stacking, which may inhibit the process of cycloaddition (*Angew. Chem. Int. Ed.* **2020**, *59*, 8828). Besides, conventional photochemistry suggests that the reaction usually occurs with atomic or molecular motions and collisions. For gel and powder samples, their tight stacking patterns restrict the spatial movement of the C=C skeleton, preventing them from obtaining ideal geometries that are conducive to [2+2] cycloaddition reactions. Therefore, the gel and supramolecular solid are basically photostable under light exposure. These above discussions have been added in the revised manuscript (Please check Page 18 for the discussion).

7. Since this study is mainly on photoluminescence. I suggest the authors demonstrate at least one set of PL spectra in the main text, NOT only CPL spectra.

Author response: We thank the reviewer's kind reminder. We have added PL spectra (Fig. 2b) in the main text.

8. The reasons for the inverted CPL are too simple and need to be discussed in detail. Will it be related to the different CH- π interaction modes in different states?

Author response: We greatly thank the reviewer's important comments. It is worth noting that many chiral luminophores fail to detect the CPL signal, even though they possess high g_{abs} . In general, the intensity and handedness of the CPL are sensitive to the chiral environment between the chromophores, which are influenced by many factors. For **S-PEB**, no CPL signal was detected in its monomeric solution (TCE solution) owing to the lack of long-range ordered

helical arrangement. For the **S-PEB** supramolecular solution, a negative CPL band with a high signal-to-noise ratio was displayed ranging from 410 to 503 nm, and a weak positive emission band was located ranging from 504 to 610 nm. This phenomenon that one compound with two different handedness of CPL signals has also been reported in previous literature (*Chem. Commun.* **2017**, 53, 6323; *Tetrahedron Lett.* **2016**, 57, 1256; *Chem. Commun.* **2014**, 50, 9951). When we switched the excitation wavelength (330 and 365 nm), their CPL profiles still had a weak positive signal at the longer wavelength, further confirming the truth of the signal. For the supramolecular gel with higher concentration, a positive CPL ($g=5 \times 10^{-3}$) was observed ranging from 430 to 630 nm, which coincides with the weak positive band in its supramolecular solution (Supplementary Fig. 21). Additionally, the drop-casting thin film and **S-PEB@PMMA** bulk film both showed similar results. Therefore, the CPL inversion is derived from the intrinsic property of **S-PEB**, such as the rotatory strength (+/-) at the varied wavelength.

We also applied theoretical calculations to simulate the CPL spectra in various states. The calculated CPL spectrum of **S-PEB** was obtained by the method of B3LYP/6-31G* (with methylcyclohexane as solvent), which shows a general validity in many previously reported chiral compounds. The CPL spectra (ECD) of one and two molecules (dimer) in methylcyclohexane (MCH) were calculated, where the two cases showed two different handedness of CPL bands in the wavelength range from 300-650 nm (Supplementary Fig. 22). Also, the CPL spectrum of **S-PEB** dimer presents an inverted handedness, ranging from 380 to 600 nm, in comparison to the corresponding single molecule. These results indicate the CPL signal of **S-PEB** is susceptible to the chromophore stacking layers, which is in good agreement with the differential CPL spectra presented by supramolecular solution and gel. In a word, we realize this color-dependent CPL via the selective supramolecular self-assembly methods. Above discussions and figures have been added to the revised manuscript (Please check Page 13-14 for the discussion, and Supplementary Fig. 21 and 22 in the Supplementary Information).

Responses to Reviewer 3:

General comments: Developing facile and efficient methods to obtain circularly polarized luminescence (CPL) materials with a large luminescence dissymmetry factor (g_{lum}) and fluorescence quantum yield (Φ_Y) is attractive but still challenging. In this nice manuscript, supramolecular polymerization of styrylpyrenes (R/S-PEB) is utilized to attain this aim, which can self-assemble into helical nanoribbons driven by unusual CH- π interactions. supramolecular solution of S-PEB shows remarkable blue-color CPL property (g_{lum} : 0.011, Φ_Y : 69%). From supramolecular solution to gel, the emission color (blue to yellow-green) and handedness of CPL (g_{lum} : - 0.011 to + 0.005) are concurrently manipulated, while the corresponding supramolecular chirality maintains unchanged, representing the rare example of color-dependent CPL materials. Benefiting from supramolecular confine effect, the [2+2] cycloaddition reaction rate of the supramolecular solution is 10.5 times higher than that of the monomeric solution. In contrast, no cycloaddition reaction occurs for the gel and assembled solid samples. Our findings provide a new vision for fabricating multi-modal and high-performance CPL-active materials, paving the way the development of advanced photo-responsive chiral devices. The work presented in this manuscript has good novelty and scientific value. The text is well organized and the results are clearly discussed. Therefore, I recommend this manuscript to be accepted after finishing the following minor revisions:

Author response: We thank the reviewer's comments and recommendation.

1. In the abstract, change "that of monomeric solution" to "that of the monomeric solution".

Author response: We thank the reviewer's kind reminder and have revised the sentence accordingly.

2. In the introduction, change "their promising application in 3D display" to "their promising applications in 3D display¹".

Author response: We thank the reviewer's kind reminder. We have changed the sentence accordingly in the revised manuscript.

3. In the introduction, change "multi-color property are rarely reported" to "multi-color property have been rarely reported"

Author response: We thank the reviewer's kind reminder. We have revised the sentence accordingly in the revised manuscript.

REVIEWERS' COMMENTS

Reviewer #1 (Remarks to the Author):

In this revised version, the authors fully addressed my earlier concerns, thus the manuscript is recommended for publication.

Reviewer #2 (Remarks to the Author):

The authors have correctly addressed all of my concerns. I can suggest its acceptance now.

Response to Reviewers' Comments

Reviewer #1 (Remarks to the Author):

In this revised version, the authors fully addressed my earlier concerns, thus the manuscript is recommended for publication.

Our response: Thanks very much for your recommendation of publication.

Reviewer #2 (Remarks to the Author):

The authors have correctly addressed all of my concerns. I can suggest its acceptance now.

Our response: Thanks very much for your recommendation of publication.